# FROM COARSE TO FINE: RECURSIVE AUDIO-VISUAL SEMANTIC ENHANCEMENT FOR SPEECH SEPARATION

## ABSTRACT

Audio-visual speech separation aims to isolate each speaker's clean voice from mixtures by leveraging visual cues such as lip movements and facial features. While visual information provides complementary semantic guidance, existing methods often underexploit its potential by relying on static visual representations. In this paper, we propose CSFNet, a Coarse-to-Separate-Fine Network that introduces a recursive semantic enhancement paradigm for more effective separation. CSFNet operates in two stages: (1) Coarse Separation, where a first-pass estimation reconstructs a coarse audio waveform from the mixture and visual input; and (2) Fine Separation, where the coarse audio is fed back into an audio-visual speech recognition (AVSR) model together with the visual stream. This recursive process produces more discriminative semantic representations, which are then used to extract refined audio. To further exploit these semantics, we design a speaker-aware perceptual fusion block to encode speaker identity across modalities, and a multi-range spectro-temporal separation network to capture both local and global time-frequency patterns. Extensive experiments on three benchmark datasets and two noisy datasets show that CSFNet achieves state-of-the-art (SOTA) performance, with substantial coarse-to-fine improvements, validating the necessity and effectiveness of our recursive semantic enhancement framework.

## 1 INTRODUCTION

Speech separation refers to the task of isolating individual speech signals from overlapping audio mixtures, which is a challenge famously exemplified by the "cocktail party problem" (Cherry, 1953), where humans demonstrate an innate ability (Conway et al., 2001; Coch et al., 2005; Mesgarani & Chang, 2012) to focus on a specific speaker amidst background noise and competing voices. While this capability comes naturally to human audition, computational approaches to speech separation have long struggled to achieve comparable performance. The advent of deep learning (Hershey et al., 2016) has opened new possibilities for tackling this problem, enabling data-driven systems to learn complex auditory patterns and achieve unprecedented separation accuracy.

Early research in this domain primarily explored audio-only methods, utilizing temporally segmented utterances such as before and after speech overlaps from multiple speakers to train separation models (Luo & Mesgarani, 2019; Luo et al., 2020; Hu et al., 2021; Wang et al., 2023; Kalkhorani et al., 2024a). To address the performance degradation caused by overlapping speech, background noise, and reverberation, subsequent work incorporated speaker enrollment information, namely voice cues from the target speaker, to guide the separation process and improve robustness (Wang et al., 2018; Xu et al., 2020; Liu et al., 2023; Xue et al., 2025).

To further improve separation performance under challenging acoustic conditions, recent studies have leveraged additional modalities such as textual (Schulze-Forster et al., 2020; Rahimi et al., 2022) and visual cues (Afouras et al., 2018c; Wu et al., 2019; Gao & Grauman, 2021; Li et al., 2024a). These approaches introduce dedicated branches for text inputs and visual signals, including lip movements and associated facial appearance features. Among available auxiliary cues like speaker enrollment, text, and video, visual information provides distinct advantages. It can be captured passively through telephoto cameras without requiring user cooperation, unlike voice or text based methods. This makes visual modality uniquely suited for practical deployment in unconstrained environments. Consequently, integrating visual information with audio representations tra-

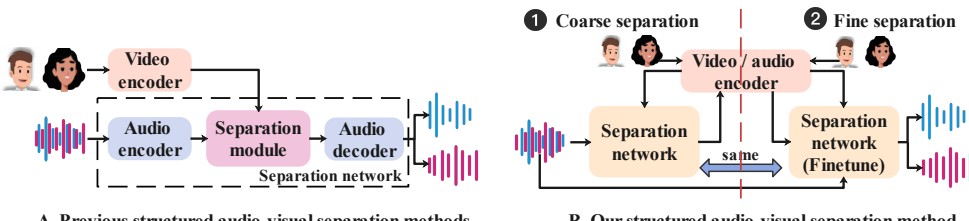

Figure 1: Previous vs. our audio-visual separation methods

ditionally used in audio-only methods has emerged as a promising direction, commonly known as audio-visual speech separation (Li et al., 2024a;b; Kalkhorani et al., 2024b; Mu & Yang, 2024).

Incorporating visual modalities brings notable gains in speech separation, mainly by exploiting semantic cues. Early works (Li et al., 2024a;b) used word-classification pretrained models, but the semantics were restricted to a 500-word vocabulary, limiting generalization. Later studies (Kalkhorani et al., 2024b; Mu & Yang, 2024) employed AVSR models (e.g., Deep-AVSR (Afouras et al., 2018b), AV-HuBERT (Shi et al., 2022)) to obtain sentence-level representations. However, these approaches still rely solely on visual input, as shown in Figure 1, which is inherently ambiguous (e.g., /p/ vs. /b/) and lacks speaker-discriminative details. As shown in Ma et al. (2023), video-only input yields much weaker representations than audio-visual joint input, leading to less effective semantics for separation, especially in multi-speaker scenarios. This motivates us to recursively integrate coarse separated audio into the AVSR model together with video, thereby extracting more discriminative and speaker-aware semantic representations.

In this work, we propose the Coarse-to-Separate-Fine Network (CSFNet), a two-stage recursive semantic enhancement framework. In *coarse separation* stage, audio mixtures and visual features are fused by a speaker-aware module and passed to a separation backbone for coarse audio estimation. In *fine separation* stage, the coarse audio and visual stream are fed into a pretrained AVSR model to obtain enhanced semantics, which are reintegrated to produce refined clean signals. To fully exploit these semantics, we design a speaker-aware perceptual fusion (SP) block to encode identity across modalities, and a multi-range spectro-temporal separation (MST) backbone to capture both local and global T-F dependencies. Our main contributions are:

1. We propose CSFNet, a coarse-to-fine framework that recursively enhances semantics by feeding coarse audio back into a AVSR model, enabling joint audio-visual semantic extraction.

2. We design a speaker-aware fusion block and a multi-range spectro-temporal network, improving robustness to modality degradation and capturing multi-scale T-F patterns.

Extensive experiments across multiple benchmarks demonstrate that CSFNet achieves new state-of-the-art performance, primarily due to the introduction of the fine separation stage. This stage significantly enhances semantic information extraction, leading to a 10% reduction in word error rate (WER) on the LRS2-2Mix dataset. Moreover, it greatly improves multi-speaker separation and maintains strong robustness even when visual information is partially or completely absent.

## 2 RELATED WORK

**Audio-Only Speech Separation**. Early research in speech separation primarily focused on classical signal processing methods such as independent component analysis (ICA) and non-negative matrix factorization (NMF). With the advent of deep learning, data-driven approaches have significantly advanced the SOTA. Representative methods include Deep Clustering (DPCL) (Hershey et al., 2016) and Deep Attractor Networks (DANet) (Chen et al., 2017), which learn discriminative embeddings for speaker assignment in the spectral domain. Subsequently, time-domain methods such as Conv-TasNet (Luo & Mesgarani, 2019) and Dual-Path RNN (DPRNN) (Luo et al., 2020) have further improved separation performance by directly modeling raw waveforms and efficiently

capturing long-range temporal dependencies. More recently, Transformer-based architectures like SepFormer (Subakan et al., 2021) have achieved SOTA performance by exploiting global contextual information across both time and frequency dimensions. In addition, complex spectral mapping networks such as TF-GridNet (Wang et al., 2023) and CrossNet (Kalkhorani & Wang, 2024) have demonstrated further improvements by explicitly modeling the complex-valued spectrum.

**Audio-Visual Speech Separation**. In recent years, incorporating visual cues into speech separation has attracted growing attention due to its robustness in noisy and overlapping speech conditions (Li et al., 2018; Rahimi et al., 2022), all of which are inherently using semantic information. Early studies including VisualVoice (Gao & Grauman, 2021) fused lip motion, facial appearance, and audio features for effective multimodal separation. Subsequent works introduced attention-based frameworks (Afouras et al., 2018c; Lin et al., 2023) to better align audio and visual streams over time. More recent approaches, inspired by the human audio-visual perception system, have integrated multi-scale features (Li et al., 2024a) or designed hierarchical fusion modules (Li et al., 2024b) to further enhance separation performance.

A notable trend on exploiting semantic information involves utilizing pretrained audio-visual speech recognition (AVSR) models, such as AV-HuBERT (Shi et al., 2022) and Deep AVSR (Afouras et al., 2018b), which have demonstrated strong performance in speech recognition tasks. To be specific, these AVSR models proposed by Shi et al. (2022) and Afouras et al. (2018b) have been employed as front-end modules to extract rich sentence-level semantics from lip movements, showing remarkable effectiveness in audio-visual speech separation (Mu & Yang, 2024; Kalkhorani et al., 2024b). However, despite these advances, a key limitation remains: relying exclusively on visual input may underutilize the semantic and speaker-discriminative information provided by the audio modality.

## 3 CSFNET

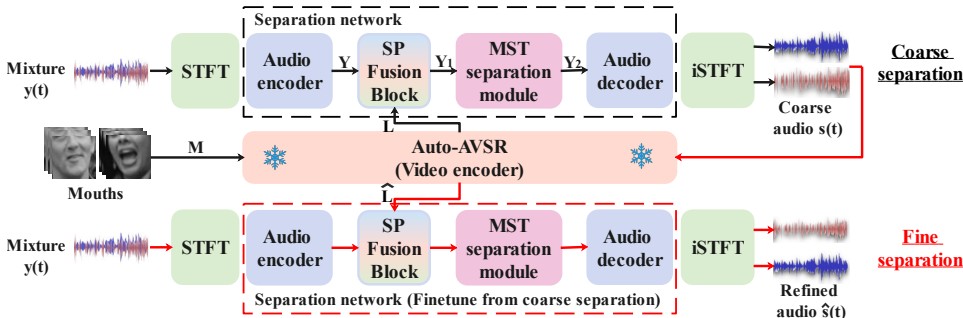

Figure 2: The overall pipeline of CSFNet comprises two stages: coarse separation and fine separation. The coarse audio from the coarse separation stage is leveraged for fine separation, where the fine separation network is finetuned based on the coarse stage to produce refined audio.

### 3.1 OVERVIEW

The CSFNet system (Figure 2) consists of five main components: an audio encoder, a video encoder (Auto-AVSR), an SP fusion block, an MST separation module, and an audio decoder. Specifically, given a mixture speech signal $y(t)$, together with the corresponding visual information of all involved speakers (i.e., extracted mouth regions, denoted as *Mouths*), the goal is to leverage this multimodal information to accurately separate and recover each speaker's voice from the mixture, even under noisy acoustic conditions. The overall process is divided into two stages. In the **coarse separation stage**, audio features $Y$ are first extracted from the mixture $y(t)$ using the audio encoder. Meanwhile, a pretrained video encoder (Auto-AVSR) produces lip-motion representations $L$. These two modalities are integrated via the SP Fusion Block to form deeply fused cross-modal representations. The MST separation module is then applied to disentangle latent speaker-specific features, and the audio decoder reconstructs the coarse speech estimates $s(t)$ for each individual speaker. In

the **fine separation stage**, the coarse estimates $s(t)$, along with the corresponding lip features $M$, are fed into the Auto-AVSR model again to obtain more accurate and discriminative semantic representations $\hat{L}$. These enhanced features are processed once more through the SP Fusion Block, MST separation module, and audio decoder, which are finetuned based on the coarse separation stage, ultimately producing refined, high-quality speech outputs $\hat{s}(t)$.

## 3.2 DETAILED ARCHITECTURE

### 3.2.1 AUDIO ENCODER

The mixed signal $y(t)$ is first transformed into a time-frequency (T-F) representation $Y_0 \in \mathbb{R}^{2 \times T \times F}$ via the Short-Time Fourier Transform (STFT). To extract informative and discriminative features from $Y_1$, we employ a multi-scale convolutional encoder that processes the input through four parallel convolutional branches with different receptive fields: a $1 \times 1$ standard convolution and three $3 \times 3$ convolutions with dilation rates $d = 1, 2, 3$. The fused output is computed as:

$$Y = \text{Concat}\left(\text{Conv}^{1 \times 1}(Y_0), \text{Conv}^{3 \times 3}_{d=1}(Y_0), \text{Conv}^{3 \times 3}_{d=2}(Y_0), \text{Conv}^{3 \times 3}_{d=3}(Y_0)\right) \tag{1}$$

where $\text{Concat}(\cdot)$ denotes channel-wise concatenation, and $\text{Conv}^{k \times k}_d(\cdot)$ indicates a 2D convolution with kernel size $k \times k$ and dilation rate $d$. The concatenated feature map is then passed through a Group Normalization layer followed by a PReLU activation function, resulting in an encoded representation $Y \in \mathbb{R}^{C \times T \times F}$.

### 3.2.2 VIDEO ENCODER (AUTO-AVSR)

We utilize a pretrained Auto-AVSR (Ma et al., 2023) as the video encoder, which consists of three components. The first is a Visual Speech Recognition (VSR) module that processes lip movement, which serves as the representation $L \in \mathbb{R}^{T_1 \times C}$ in the first coarse separation stage. The second is an Audio Speech Recognition (ASR) module that processes the audio from the first stage output. The final is a Classic Multilayer Perceptron (MLP), which takes the representation in the first stage and combines it with the audio features output from the ASR module to form $\hat{L} \in \mathbb{R}^{T_1 \times C}$. For more details, please refer to Appendix B.

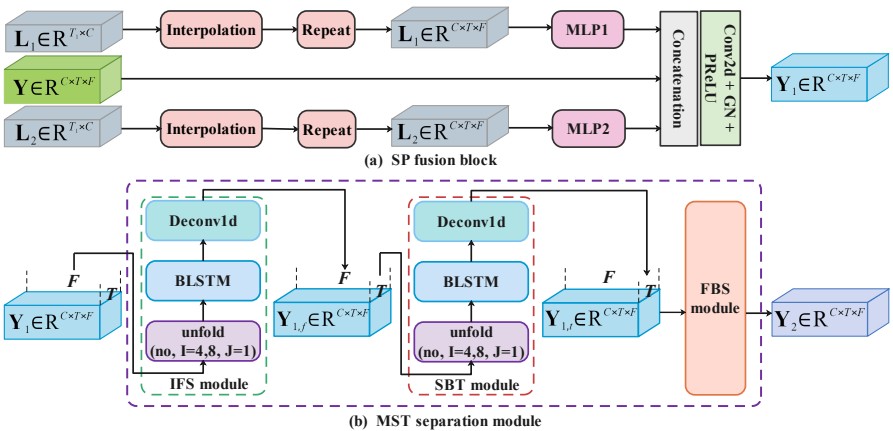

(a) SP fusion block

(b) MST separation module

Figure 3: The two main components of CSFNet: (a) SP fusion block, (b) MST separation module.

### 3.2.3 SPEAKER-WISE PERCEPTUAL (SP) FUSION BLOCK

Unlike previous works (Lee et al., 2021; Lin et al., 2023; Liu et al., 2024; Kalkhorani et al., 2024a) that rely on complex attention mechanisms, we propose a simple yet efficient fusion module named *Speaker-wise Perceptual (SP) Fusion Block*. This module is designed to enhance the representation of mixed speech by incorporating both speaker-specific and global contextual information. Given the mixed audio feature $Y$, we perform speaker-wise fusion with the corresponding visual features.

For each speaker in the mixture, we obtain a lip movement feature $L_i \in \mathbb{R}^{T_1 \times C}$. Taking a 2-speaker mixture as an example (as illustrated in Figure 3-a), to align $L_i$ with the $Y$ in both temporal and frequency dimensions, we first apply *linear interpolation* to upsample the temporal dimension $T_1$ to match the number of audio frames $T$. Then, the visual features are expanded along the frequency axis by repeating each time step across all frequency bins, forming $L_i \in \mathbb{R}^{C \times T \times F}$. We then fuse $Y$ with each pair of visual features $L_1, L_2$ separately using a simple multilayer perceptron (MLP). This results in three cross-modal feature representations, which are then concatenated together with the original audio feature $Y$. The concatenated feature is subsequently passed through a lightweight convolutional block consisting of a 2D convolution layer, Group Normalization, and a PReLU activation function to obtain the final fused representation $Y_1 \in \mathbb{R}^{C \times T \times F}$. This design allows the model to capture both fine-grained speaker-specific cues and global contextual information from the mixture. Moreover, as evidenced by our experiments, the proposed fusion block exhibits strong generalization ability, even when visual inputs from one speakers are missing.

### 3.2.4 Multi-range Spectro-Temporal (MST) separation module

Our separation backbone builds on TF-GridNet (Wang et al., 2023), which comprises three modules: an Intra-Frame Spectral (IFS) module, a Sub-Band Temporal (SBT) module, and a Full-Band Self-Attention (FBS) module. However, its use of a single `unfold` range and a large BLSTM hidden size ($H = 256$) limits the ability to capture multi-scale spectro-temporal features and incurs high computational cost. To address this, we propose the Multi-range Spectro-Temporal (MST) module (as shown in Figure 3-b).

In MST, both spectral and temporal modules adopt a parallel multi-branch design: (1) a global branch (no `unfold`), (2) a medium-range branch ($I = 4$, $J = 1$), and (3) a large-range branch ($I = 8$, $J = 1$). The outputs from all branches are concatenated along the channel dimension and subsequently processed by a 2D convolution layer, producing richer and more informative multi-scale representations, improving performance in overlapping-speaker scenarios. To enhance efficiency, we reduce the BLSTM hidden size to $H = 96$. Further ablation details can be found in Appendix C. For the FBS module, we retain the multi-head attention structure from (Wang et al., 2023) with $N$ heads. In practice, $B$ MST blocks are stacked to form the separation backbone, yielding the final output $Y_2 \in \mathbb{R}^{C \times T \times F}$.

### 3.2.5 Audio decoder

As for the decoder, a 2D transposed convolution is employed to transform the separated features into time-frequency (T-F) spectrograms corresponding to each speaker. These spectrograms are then converted into time-domain waveforms through the inverse short-time Fourier transform (iSTFT), yielding the final separated speech signal $\hat{s}(t)$.

## 4 Experiments and Results

### 4.1 Datasets

We conducted experiments using a variety of publicly available audiovisual datasets under both clean and noisy conditions. For clean conditions, we used LRS2 (Afouras et al., 2018b), LRS3 (Afouras et al., 2018a), and VoxCeleb2 (Chung et al., 2018), from which we created multi-speaker mixtures of 2–4 speakers with SNRs in $[-5, 5]$ dB. For noisy conditions, we employed NTCD-TIMIT (Abdelaziz et al., 2017) and LRS3+WHAM! (Wichern et al., 2019), generating mixtures with two speakers and background noise at varying SNRs. Visual inputs were preprocessed following prior work (Ma et al., 2021; 2023), extracting mouth ROIs from video frames at 25 FPS and resizing to $88 \times 88$ pixels. Detailed descriptions of dataset partitioning, mixture generation, and visual preprocessing are provided in Appendix D.

### 4.2 Implementation Details

In our proposed CSFNet framework, the STFT block employs a square-root Hann window with a window size of 32 ms and a hop size of 8 ms, resulting in 257 frequency bins ($F = 257$). The audio-only model is implemented by simply removing the SP fusion block along with the corresponding

| Method | Params (M) | Mod. | LRS2 | | LRS3 | | VoxCeleb2 | |
|---|---|---|---|---|---|---|---|---|
| | | | SI-SDRi ↑ | SDRi ↑ | SI-SDRi ↑ | SDRi ↑ | SI-SDRi ↑ | SDRi ↑ |
| Unprocessed | – | – | 0.0 | 0.0 | 0.0 | 0.0 | 0.0 | 0.0 |
| DPCL++ (Hershey et al., 2016) | 13.6 | A | 3.3 | 4.3 | 5.8 | 6.2 | 2.1 | 2.5 |
| Conv-TasNet (Luo & Mesgarani, 2019) | 5.6 | A | 10.3 | 10.7 | 11.1 | 11.4 | 6.9 | 7.5 |
| SuDoRM-RF (Tzinis et al., 2020) | 2.7 | A | 9.1 | 9.5 | 12.1 | 12.3 | 6.5 | 6.9 |
| A-FRCNN (Hu et al., 2021) | 6.3 | A | 9.4 | 10.1 | 12.5 | 12.8 | 7.8 | 8.2 |
| TF-GridNet (Wang et al., 2023) | 14.5 | A | 13.9* | 14.0* | 17.3* | 17.4* | 11.2* | 11.5* |
| Crossnet (Kalkhorani & Wang, 2024) | 6.6 | A | 14.1* | 14.3* | 17.2* | 17.4* | 11.4* | 11.7* |
| **CSFNet (Audio-Only)** | 10.7 | A | **14.5** | **14.6** | **17.4** | **17.7** | **11.9** | **12.5** |
| The Conversation (Afouras et al., 2018c) | 62.7 | AV | – | – | – | – | – | 8.9 |
| VisualVoice (Gao & Grauman, 2021) | 77.8 | AV | 11.5 | 11.8 | 9.9 | 10.3 | 9.3 | 10.2 |
| AVConvTasNet (Wu et al., 2019) | 16.5 | AV | 12.5 | 12.8 | 11.2 | 11.7 | 9.2 | 9.8 |
| AVLiT (Martel et al., 2023) | 5.8 | AV | 12.8 | 13.1 | 13.5 | 13.6 | 9.4 | 9.9 |
| CTCNet (Li et al., 2024a) | 7.0 | AV | 14.3 | 14.6 | 17.4 | 17.5 | 11.9 | 13.1 |
| RTFS-Net (Pegg et al., 2023) | 0.7 | AV | 14.9 | 15.1 | 17.5 | 17.6 | 12.4 | 13.6 |
| AVSepChain (Mu & Yang, 2024) | 33.1 | AV | 15.3 | 15.7 | – | – | 13.6 | 14.2 |
| IIANet (Li et al., 2024b) | 3.1 | AV | 16.0 | 16.2 | 18.3 | 18.5 | 13.6 | 14.3 |
| **CSFNet (fine separation)** | 10.9 | AV | **16.8** | **16.9** | **19.4** | **19.7** | **14.8** | **15.1** |
| AV-CrossNet (Kalkhorani et al., 2024b) (DM) | 11.1 | AV | 16.8 | 17.1 | 18.3 | 18.5 | 14.6 | **14.9** |
| **CSFNet (DM)(coarse separation)** | 10.9 | AV | **16.9** | **17.0** | **18.9** | **19.2** | **14.8** | **14.9** |
| **CSFNet (DM)(fine separation)** | 10.9 | AV | **17.5** | **17.8** | **20.1** | **20.2** | **15.4** | **15.6** |

Table 1: Speaker separation results of different AVSS methods on LRS2, LRS3, and VoxCeleb2 datasets. 'Mod.' stands for modality (audio-only (A) or audio-visual (AV)). DM indicates dynamic mixing conditions. * denotes results reproduced by our implementation.

| Method | Pretrain Model | SI-SDRi | Computation Cost | | GPU Metrics | |
|---|---|---|---|---|---|---|
| | | | MACs (G) | Params (M) | Time (ms) | Memory (MB) |
| VisualVoice (Gao & Grauman, 2021) | Yes | 11.5 | **9.7** | 77.8 | 110.31 | 313.74 |
| AVConvTasNet (Wu et al., 2019) | No | 12.5 | 23.8 | 16.5 | **62.51** | 117.05 |
| AVLiT (Martel et al., 2023) | Yes | 12.8 | 18.2 | 5.8 | **62.51** | 24.00 |
| CTCNet (Li et al., 2024a) | Yes | 14.3 | 167.1 | 7.0 | 84.17 | 75.80 |
| IIANet (Li et al., 2024b) | Yes | 16.0 | 18.6 | **3.1** | 110.11 | **12.50** |
| AV-CrossNet (Kalkhorani et al., 2024b) | Yes | 16.8 | 92.3 | 11.1 | 367.91 | 40.5 |
| **CSFNet(coarse separation)** | Yes | **16.9** | 31.7 | 10.9 | 117.39 | 42.5 |
| **CSFNet(fine separation)** | Yes | **17.5** | 64.8 | 10.9 | 228.15 | 83.5 |

Table 2: Computational complexities of different AVSS methods. All results are measured with an input audio length of 1 second, a sample rate of 16 kHz, and a video frame sample rate of 25 FPS.

visual inputs, which reduces the total number of parameters by approximately 0.2M. The feature channel dimension of audio encoder is set to $C = 192$. The number of self-attention heads is set to $N = 4$, and the number of MST separation blocks is $B = 6$. We use RetinaFace (Deng et al., 2019) as the mouth region detector. The model is optimized using the Adam optimizer. We use a batch size of 16 for coarse separation stage and 8 for fine separation stage. The initial learning rate is set to 0.001 for coarse separation stage and 0.0001 for fine separation stage. In both stages, we employ a ReduceLROnPlateau scheduler with a patience of 3, meaning the learning rate is halved if the validation loss does not improve for three consecutive epochs. The maximum number of training epochs is 200, and gradient clipping is applied with a maximum norm of 5. With regard to the loss function, we employ a combination of magnitude and scale-invariant signal-to-distortion ratio (SI-SDR) loss, which is similar to AV-Crossnet (Kalkhorani et al., 2024b). See Appendix E for more details. All experiments are conducted using 8 NVIDIA A100 GPUs (40 GB).

### 4.3 EVALUATION METRICS

The separated signal performance metrics include scale-invariant signal-to-distortion ratio improvement (SI-SDRi) (Le Roux et al., 2019), signal-to-distortion ratio improvement (SDRi) (Vincent et al., 2006), perceptual evaluation of speech quality (PESQ) (Rix et al., 2001) and extended short-time objective intelligibility (eSTOI) (Jensen & Taal, 2016). Moreover, we also measure the word error rate (WER), where a lower error rate indicates better preservation and restoration of the content information in the speech. We report parameters and MAC operations for complexity, which are calculated for one second of audio at 16 kHz. Inference speed was measured on NVIDIA A100.

| Method | Mod. | NTCD-TIMIT | | | LRS3+WHAM! | | |
|--------|------|-------|-------|---------|-------|-------|---------|
| | | PESQ↑ | eSTOI↑ | SI-SDRi↑ | PESQ↑ | eSTOI↑ | SI-SDRi↑ |
| Unprocessed | – | 1.19 | 0.33 | – | 1.08 | 0.37 | – |
| ConvTasNet (Luo & Mesgarani, 2019) | A | 1.35 | 0.38 | 8.76 | 1.24 | 0.51 | 9.66 |
| A-FRCNN (Hu et al., 2021) | A | 1.30 | 0.31 | 6.92 | 1.25 | 0.49 | 9.21 |
| DPRNN (Luo et al., 2020) | A | 1.32 | 0.39 | 9.31 | **1.40** | 0.45 | **10.93** |
| **CSFNet (Audio-only)** | A | **1.78** | **0.43** | **10.55** | 1.33 | **0.55** | 10.91 |
| L2L (Ephrat et al., 2018) | AV | 1.23 | 0.26 | 3.36 | 1.16 | 0.51 | 7.60 |
| LAVSE (Chuang et al., 2020) | AV | 1.31 | 0.37 | 6.22 | 1.24 | 0.50 | 5.59 |
| AVConvTasNet (Wu et al., 2019) | AV | 1.33 | 0.40 | 9.02 | 1.29 | 0.60 | 6.21 |
| VisualVoice (Gao & Grauman, 2021) | AV | 1.45 | 0.43 | 10.04 | 1.48 | 0.63 | 11.87 |
| AVLiT (Martel et al., 2023) | AV | 1.43 | 0.45 | 11.00 | 1.52 | 0.68 | 12.42 |
| IIANet (Li et al., 2024b) | AV | 2.00 | 0.49 | 13.56 | 1.55 | 0.68 | 11.93 |
| **CSFNet (coarse separation)** | AV | **2.02** | **0.51** | **13.74** | **2.42** | **0.85** | **13.05** |
| **CSFNet (fine separation)** | AV | **2.15** | **0.61** | **15.67** | **2.72** | **0.91** | **14.71** |

Table 3: Separation results on noisy NTCD-TIMIT and LRS3+WHAM! datasets. Comparison results are from (Martel et al., 2023).

## 4.4 COMPARISON WITH THE STATE-OF-THE-ART (SOTA)

### 4.4.1 AUDIOVISUAL SPEAKER SEPARATION IN CLEAN CONDITIONS

Table 1 presents a comprehensive comparison between our proposed CSFNet and existing state-of-the-art methods on the LRS2, LRS3, and VoxCeleb2 datasets. The benchmarks are categorized into two groups. The first group includes *audio-only* approaches that do not leverage any visual information. Based on our reproduced results, we observe that MAVINet still delivers gains over strong audio-only baselines. The second group consists of *audio-visual* methods that incorporate visual cues. In the first stage, CSFNet achieves only marginal improvements over the SOTA models, with an increase of around 0.2 dB. However, in the second fine separation stage, CSFNet brings substantial performance gains compared to the coarse separation stage, achieving improvements of 0.6 dB, 0.8 dB, and 0.7 dB on LRS2, LRS3, and VoxCeleb2, respectively, thereby demonstrating the effectiveness of the refinement process. Notably, under the dynamic mixing (DM) setting which generates mixtures on the fly during training with varying speakers and SNRs, our model achieves a further 0.7 dB SI-SDR improvement over the previous best, highlighting both its robustness and superiority.

### 4.4.2 COMPUTATIONAL COST OF AVSS MODELS

Table 2 reports the computational complexity of different AVSS models. As shown, most approaches rely on pretrained models, which do not introduce a significant increase in computational cost while enhancing separation performance. Compared with all methods that incorporate pretrained models, our CSFNet remains on the same order of magnitude in terms of MACs, parameter size, GPU inference time, and memory consumption. Moreover, relative to IIANet, CSFNet requires roughly three times more parameters and MACs, yet achieves a substantial improvement in SI-SDR, clearly demonstrating the efficiency performance trade-off of our design.

### 4.4.3 AUDIOVISUAL SPEAKER SEPARATION IN NOISY CONDITIONS

Table 3 presents a comparison between our model, CSFNet, and the best-performing baseline models under more challenging noisy conditions, covering both audio-only and audio-visual approaches. The evaluation is conducted on two benchmark datasets: NTCD-TIMIT and LRS3+WHAM!, using PESQ, eSTOI, and SI-SDRi as the evaluation metrics. It is evident that our model consistently achieves the best results across all three metrics on both datasets. On NTCD-TIMIT, CSFNet shows a particularly notable improvement in SI-SDRi with a relative gain. On the more challenging LRS3+WHAM! dataset, it outperforms the SOTA AVLiT model by roughly 2 dB in SI-SDRi. Furthermore, even in the audio-only setting, CSFNet achieves comparative performance upon the strong audio-only model DPRNN. These consistent improvements across different settings and datasets

| Method | LRS2-2Mix | | LRS2-3Mix | | LRS2-4Mix | |
|---|---|---|---|---|---|---|
| | SI-SNRi↑ | SDRi↑ | SI-SNRi↑ | SDRi↑ | SI-SNRi↑ | SDRi↑ |
| AVConvTasNet (Wu et al., 2019) | 12.5 | 12.8 | 8.2 | 8.8 | 4.1 | 4.6 |
| AVLiT (Martel et al., 2023) | 12.8 | 13.1 | 9.4 | 9.9 | 5.0 | 5.7 |
| CTCNet (Li et al., 2024a) | 14.3 | 14.6 | 10.3 | 10.8 | 6.3 | 6.9 |
| IIANet (Li et al., 2024b) | 16.0 | 16.2 | 12.6 | 13.1 | 7.8 | 8.3 |
| **CSFNet (coarse separation)** | **16.2** | **16.3** | **13.3** | **13.5** | **10.5** | **10.8** |
| **CSFNet (fine separation)** | **16.8** | **16.9** | **15.4** | **15.5** | **14.3** | **14.5** |

Table 4: Performance comparison between the coarse and fine separation stages with varying numbers of speakers on the LRS2 dataset.

| Method | Coarse separation | | Fine separation | |
|---|---|---|---|---|
| | Input WER(%)(V) | Output WER(%)(A) | Input WER(%)(AV) | Output WER(%)(A) |
| CTCNet-Lip (Li et al., 2024a) | - | 25.03 | - | - |
| Deep-AVSR (Afouras et al., 2018b) | 58.81 | 24.94 | 23.99 | 21.78 |
| AV-HuBERT (Shi et al., 2022) | 44.42 | 24.66 | 22.87 | 21.50 |
| **Auto-AVSR (Ma et al., 2023)** | **30.67** | **22.75** | **20.03** | **18.89** |

Table 5: WER reduction on different visual frontends and AVSR models in a two-stage framework on LRS2-2Mix. "Input WER (%)" indicates the word error rate from the input. "Output WER (%)" measures the word error rate of the separated audio. All evaluations are based on 2s segments.

demonstrate the effectiveness and robustness of CSFNet for speech separation in complex acoustic environments.

## 4.5 ABLATION STUDY

### 4.5.1 IMPORTANCE OF TWO-STAGE SEPARATION

**Multi-speaker separation**. We report the results of both coarse separation stage and fine separation stage on the LRS2 dataset under 2-, 3-, and 4-speaker mixture conditions, as shown in Table 4. Notably, our model slightly outperforms the SOTA IIANet in the coarse separation stage. In the fine separation stage, the performance is further boosted, with gains of 0.6 dB, 2.1 dB, and 3.8 dB for the 2Mix, 3Mix, and 4Mix settings, respectively. These results show that with more overlapping speakers, separation becomes more challenging, highlighting the need for finer-grained semantic representations and confirming the effectiveness of the second stage.

**WER reduction**. We evaluate WER across both stages against representative models, including CTCNet-Lip (Li et al., 2024a), AV-HuBERT, and Deep-AVSR (as in AV-CrossNet). As shown in Table 5, two key findings emerge. First, while the coarse separation stage substantially reduces WER compared to the mixed input, the fine separation stage further enhances semantic information through audio-visual fusion, achieving the lowest WER overall and confirming its effectiveness. Second, CTCNet-Lip performs worst as it targets isolated word prediction, whereas Auto-AVSR yields the best visual-only WER and strongest separation among the remaining models, underscoring the crucial role of accurate semantic cues.

**Visual occlusion**. To further validate the effectiveness of our two-stage framework, we evaluate the model under more realistic conditions where visual information may be degraded or partially missing, such as in the presence of motion blur, low lighting, or severe occlusions. In such cases, we consider the visual information in the affected frames to be missing or unreliable. Specifically, we consider two scenarios: (1) only one of the two speakers gradually loses visual cues, from 0 to all frames missing; and (2) both speakers gradually lose visual cues, also from 0 to all frames missing. The data generation procedure follows Appendix F. As shown in Figure 4, missing visual cues substantially degrade model performance, particularly when more than half of the frames are lost. Nevertheless, across all cases, the second refinement stage consistently outperforms the coarse-only stage. More importantly, when more than half of the visual frames are missing, the performance

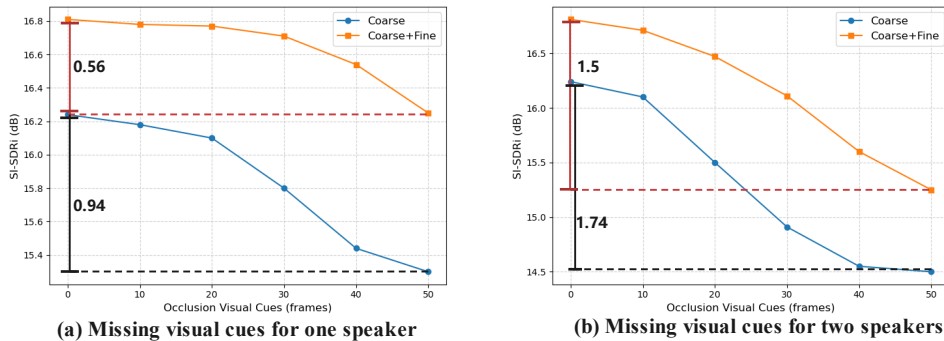

(a) **Missing visual cues for one speaker**
(b) **Missing visual cues for two speakers**

Figure 4: SI-SDRi under different numbers of missing visual cue frames for (a) one speaker, (b) two speakers on the LRS2-2Mix dataset.

drop with the two-stage framework is noticeably smaller. This suggests that the refinement stage benefits from the coarse separation outputs, which provide complementary semantic information, thereby mitigating the impact of severe visual occlusion.

### 4.5.2 DIFFERENT FUSION STRATEGIES

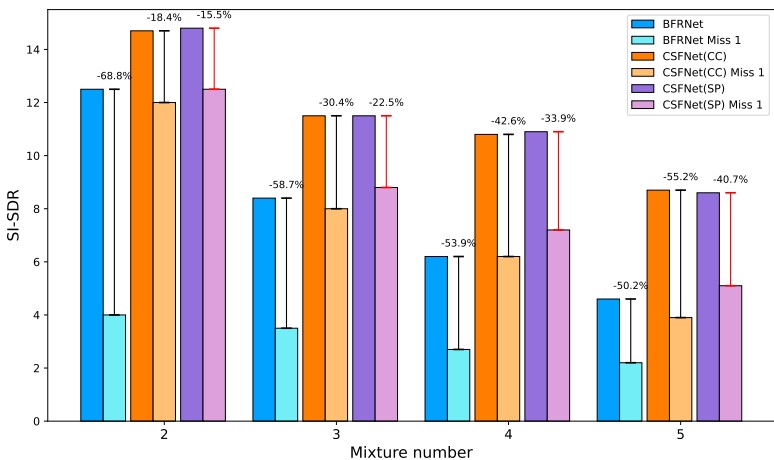

Figure 5: Comparison of fusion strategies (SP vs. CC) and audio-visual separation methods under different mixing conditions on VoxCeleb2. SP denotes the speaker-wise perceptual fusion block, CC the simple concatenation, and "Miss 1" indicates that one speaker's visual stream is missing. When a larger portion of visual input is absent, the advantage of any fusion strategy diminishes, rendering them less meaningful.

To evaluate the robustness of our fusion design, we conducted ablation experiments by (1) replacing SP fusion with simple concatenation and (2) comparing against SOTA methods, including BFRNet (Cheng et al., 2023). As shown in Figure 5, our model consistently achieves the best performance across 2–5 speaker mixtures. Furthermore, while all methods degrade under missing visual input, our approach exhibits the most graceful decline. Notably, SP fusion clearly outperforms simple concatenation, validating the effectiveness and robustness of the proposed strategy.

## 5 CONCLUSION

In this paper, we propose CSFNet, a Coarse-to-Separate-Fine network that leverages visual semantic cues for speech separation through coarse audio reconstruction and AVSR-guided refinement.

By further incorporating speaker-wise perceptual fusion and multi-range spectro-temporal modeling, CSFNet effectively encodes speaker identity across modalities and captures multi-scale time-frequency patterns. As a result, it achieves SOTA performance across multiple clean and noisy benchmarks, demonstrating both its robustness in complex acoustic conditions and its adaptability to real-world multi-speaker scenarios.

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

## A    LLM USAGE

Regarding the use of large language models (LLMs), we employ them solely for polishing the writing. Specifically, after completing all of our original drafting, we use LLMs to refine and enhance the clarity, fluency, and professional presentation of our text.

## B    AUTO-AVSR

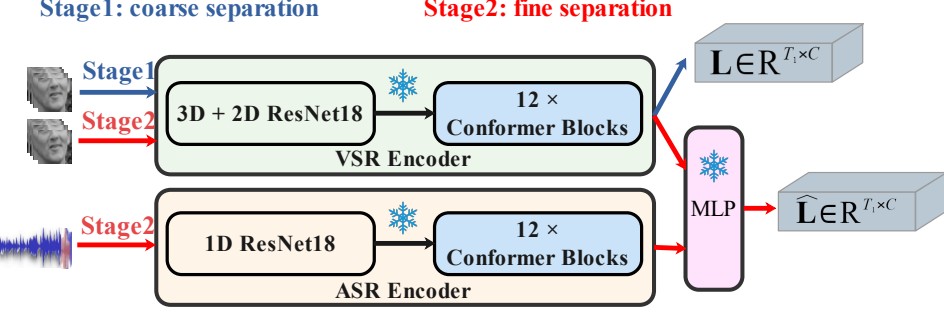

Figure 6: Detailed Flowchart of the Video Encoder (Auto-AVSR)

Figure 6 illustrates the detailed flowchart of Auto-AVSR, our visual encoder. The encoder is utilized in both stages of the separation process. In the first coarse separation stage, only lip movements are used as input, which are processed by the VSR encoder. The VSR encoder consists of a ResNet-18 backbone followed by 12 Conformer blocks, producing the stage-one semantic representation $L \in \mathbb{R}^{T_1 \times C}$. In the second fine separation stage, the coarse audio output from the first stage and the lip movements are both used as inputs. They are processed separately by the VSR encoder and the ASR encoder, respectively. The outputs of both encoders are then fed into a pre-trained MLP layer to generate the stage-two semantic representation $\hat{L} \in \mathbb{R}^{T_1 \times C}$, which is richer and more discriminative.

# C   ABLATION STUDY ON SEPARATION MODULE

| Structure | Configuration | Params (M) | SI-SNRi (dB) | SDRi (dB) | MACs (G) | GPU Time (ms) |
|---|---|---|---|---|---|---|
| Dual Path Transformer (DPT) | - | 2.9 | 15.1 | 15.3 | - | - |
| TF-GridNet (base) | H=256 | 14.5 | 16.2 | 16.4 | 78.8 | 136.45 |
| MST | H=64 | 7.3 | 16.3 | 16.4 | 49.8 | 96.77 |
| **MST** | **H=96** | **10.9** | **16.8** | **16.9** | **64.8** | **117.39** |
| MST | H=128 | 15.6 | 16.8 | 17.1 | 80.0 | 165.99 |
| MST | H=192 | 26.3 | 17.1 | 17.3 | 110.3 | 2163.17 |

Table 6: Ablation study on different separation blocks.

In our Separation Module (MST), to validate the rationale behind our hyperparameter settings, we conducted experiments using dual path transformer and the original TF-GridNet separation module as well as our proposed multi-range `unfold` design, as shown in Figure 7, with different hidden dimensions of BLSTM.

As shown in Table 6, compared with TF-GridNet (base), DPT has relatively fewer parameters and a simpler computational complexity, but its performance is also the lowest. In contrast, for our proposed MST, increasing the hidden dimension $H$ leads to a larger number of parameters and improved performance. However, given the substantial increase in computational cost, the performance gains are relatively marginal. Therefore, we select $H = 96$ as the optimal hyperparameter, which results in fewer parameters than the base model while achieving an approximately 0.6 dB improvement in performance.

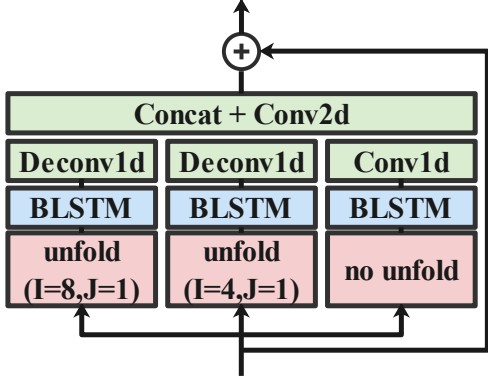

Figure 7: The details of multi-range unfold design

# D   DATASET DETAILS

In this appendix, we provide a detailed description of the datasets used in our experiments, including the procedures for creating multi-speaker mixtures and the preprocessing of visual inputs. The datasets span both clean and noisy conditions, and contain diverse audiovisual content to evaluate model robustness.

## D.1   CLEAN CONDITIONS

For clean conditions, we employed three publicly available datasets: LRS2 (Afouras et al., 2018b), LRS3 (Afouras et al., 2018a), and VoxCeleb2 (Chung et al., 2018).

**LRS2 and VoxCeleb2** are collected from YouTube videos, featuring diverse and acoustically complex environments. This diversity presents greater challenges for model generalization and robustness due to varying recording conditions, background sounds, and speaker demographics.

**LRS3** consists primarily of TED and TEDx talks, providing long, naturally spoken, and coherent sentences. This dataset enables evaluation on more structured speech, complementing the diversity found in LRS2 and VoxCeleb2.

For all three datasets, audio samples were segmented into 2-second clips with a sampling rate of 16 kHz, following the settings of previous studies. Multi-speaker mixtures were then generated by randomly selecting 2, 3, or 4 speakers from each dataset. The selected utterances were mixed with a signal-to-noise ratio (SNR) uniformly sampled in the range of $[-5, 5]$ dB to simulate realistic overlapping speech scenarios.

## D.2 NOISY CONDITIONS

To evaluate model performance under challenging acoustic environments, we utilized NTCD-TIMIT (Abdelaziz et al., 2017) and the LRS3+WHAM! dataset.

**NTCD-TIMIT** was originally designed for audiovisual noisy speech recognition with single speakers. To simulate multi-speaker mixtures, we randomly selected utterances from two different speakers, ensuring no overlap in either speaker identity or spoken content. Background noise was then added to the mixtures, with the noise SNR uniformly sampled in the range of $[-5, 20]$ dB, reflecting a wide spectrum of real-world noise conditions.

**LRS3+WHAM!** combines audiovisual speech data from LRS3 (Afouras et al., 2018a) with real-world background noise from the WHAM! dataset Wichern et al. (2019). Two-speaker mixtures were synthesized with background noise added, and the clean speech SNRs were uniformly sampled in the range of $[-5, 5]$ dB.

Across all datasets, test speakers are strictly disjoint from the training and validation sets to ensure speaker independence and fair evaluation. Dataset partitioning followed the protocols established in prior work Pegg et al. (2024); Li et al. (2024b), facilitating comparison with existing methods.

## D.3 VISUAL MODALITY PREPROCESSING

Visual data preprocessing followed the pipelines adopted in prior studies (Ma et al., 2021; 2023). Lip frames were synchronized with the audio at 25 FPS. The mouth region of interest (ROI) was extracted using a bounding box of size $96 \times 96$ pixels. To align with the input requirements of pretrained visual feature extractors used in our experiments, center cropping was applied to obtain a final input size of $88 \times 88$ pixels. This preprocessing ensures consistency across datasets and facilitates effective audiovisual feature extraction for downstream tasks.

# E LOSS FUNCTION

We optimize our model using a combination of the magnitude loss, $L_{\text{Mag}}$, and the scale-invariant signal-to-distortion ratio (SI-SDR) loss, $L_{\text{SI-SDR}}$. The SI-SDR is computed in its standard form, where the target signal is appropriately scaled to match the amplitude of the estimated signal. Additionally, the magnitude loss is normalized by the L1 norm of the target signal's magnitude in the STFT domain to ensure consistent scaling.

The loss functions are formally defined as follows:

$$L = L_{\text{Mag}} + L_{\text{SI-SDR}}, \tag{2}$$

$$L_{\text{Mag}} = \frac{\big\| |\text{STFT}(\hat{s}_c)| - |\text{STFT}(s_c)| \big\|_1}{\big\| |\text{STFT}(s_c)| \big\|_1}, \tag{3}$$

$$L_{\text{SI-SDR}} = -\frac{1}{C} \sum_{c=1}^{C} 10 \log_{10} \frac{\|s_c\|_2^2}{\|\hat{s}_c - \alpha_c s_c\|_2^2}, \tag{4}$$

$$\alpha_c = \frac{s_c^T \hat{s}_c}{s_c^T s_c}. \tag{5}$$

Here, $\|\cdot\|_1$ denotes the L1 norm, $|\cdot|$ represents the magnitude operator, $\alpha_c$ is the scaling factor, and $(\cdot)^T$ indicates the transpose. During AVSS training, permutation invariant training (PIT) is applied to resolve permutation ambiguity, ensuring that each estimated signal is consistently aligned with its corresponding target.

## F    DATASET FOR MISSING VISUAL CUES GENERATION

We follow prior works and process the video input at 25 frames per second over a 2-second window, resulting in 50 frames per sample. To simulate partial visual occlusion in the two-speaker mixed data, we consider two scenarios: (1) Only one speaker's visual input is partially occluded. We randomly remove a consecutive block of 5, 10, 20, 30, 40, or all frames. For example, in the case of 10 missing frames, we randomly choose a starting position and replace the subsequent 10 frames with zero-valued frames. This operation ensures that the temporal structure and frame alignment of the sequence are preserved despite the simulated occlusion. (2) Both speakers' visual inputs are partially occluded. Each speaker has the same number of missing frames overall, but the occluded segments are applied independently and do not occur at the same timestamps. For instance, for 10 missing frames, each speaker loses a random consecutive block of 10 frames at different positions. As in the first scenario, all removed frames are replaced by zero-valued frames so that the overall temporal continuity and alignment of the video streams remain intact.

## G    FACE ENCODER

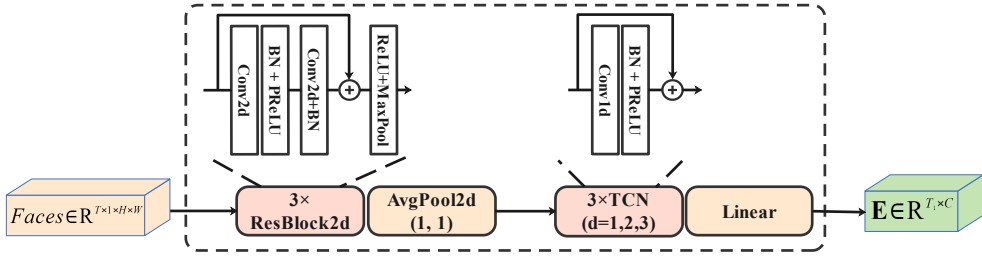

Figure 8: The details of our designed face encoder.

To verify that the second stage of our model can extract more **discriminative and speaker-aware semantic representations**, we design a representative experiment based on a face encoder. Specifically, we develop a lightweight face encoder that processes grayscale face image sequences, as illustrated in Figure 8. The visual input Faces $\in \mathbb{R}^{T \times 1 \times H \times W}$ is first passed through three stacked 2D residual convolutional blocks. Each block contains two convolutional layers with a $3 \times 3$ kernel, followed by batch normalization and PReLU activations. Downsampling is performed using max pooling with a stride of 2, and a $1 \times 1$ convolution is employed to align residual connections whenever the input and output dimensions differ. The extracted spatial features are subsequently fed

into a three-layer Temporal Convolutional Network (TCN) with increasing dilation factors $(1, 2, 4)$ to effectively capture temporal dependencies. Finally, a linear projection layer maps the output into a fixed-dimensional lip embedding vector $E \in \mathbb{R}^{T_1 \times C}$, which is then temporally interpolated to length $T$, expanded along the frequency dimension $F$, and fused with the speech features through the SP fusion block.

| Method | Params(M) | Coarse separation | | | Fine separation | | |
| --- | --- | --- | --- | --- | --- | --- | --- |
| | | SI-SDRi↑ | SDRi↑ | PESQ↑ | SI-SDRi↑ | SDRi↑ | PESQ↑ |
| CSFNet (Lip+face) | 13.4 | 16.5 | 16.6 | 3.40 | 16.8 | 16.9 | 3.46 |
| **CSFNet (Lip-only)** | **10.9** | **16.2** | **16.3** | **3.37** | **16.8** | **16.9** | **3.45** |

Table 7: Ablation study on lip-only and lip+face model performance on LRS2-2Mix.

Table 7 presents the separation results obtained when using only lip information and when using both lip and face information. It is evident that in the first coarse separation stage, the combination of lip and face information outperforms the lip-only setting. This can be attributed to the fact that lip features primarily encode semantic information but lack personalized cues related to speaker identity, while the face encoder provides complementary identity-related information such as gender. However, in the second fine separation stage, the performance of the lip+face setting becomes comparable to that of lip-only. We hypothesize that this is because the introduction of personalized speech representations in the second stage already enables the model to extract more discriminative and speaker-aware semantic representations, thereby diminishing the additional benefit of face information. Consequently, our model can achieve the benefits of incorporating face information without explicitly relying on a face encoder in the final experiments.

## H  VISUALIZATION

To more intuitively demonstrate the performance improvement achieved by the fine separation stage , we provide the following visual examples. The spectrograms below (Figure 9) illustrate the outputs of our proposed CSFNet model at both its first (coarse separation) and second (fine separation) stages on the LRS2-4Mix datasets. From left to right, each set of spectrograms corresponds to: the ground-truth audio, the output from the first (coarse) stage, and the output from the second (fine) stage. As clearly shown, the coarse separation stage fails to preserve many spectral details. In contrast, the fine separation stage of CSFNet successfully recovers and reconstructs most of the missing spectral features across almost all frequency bands. These results strongly validate the indispensability of the fine separation stage in enhancing speech separation quality.

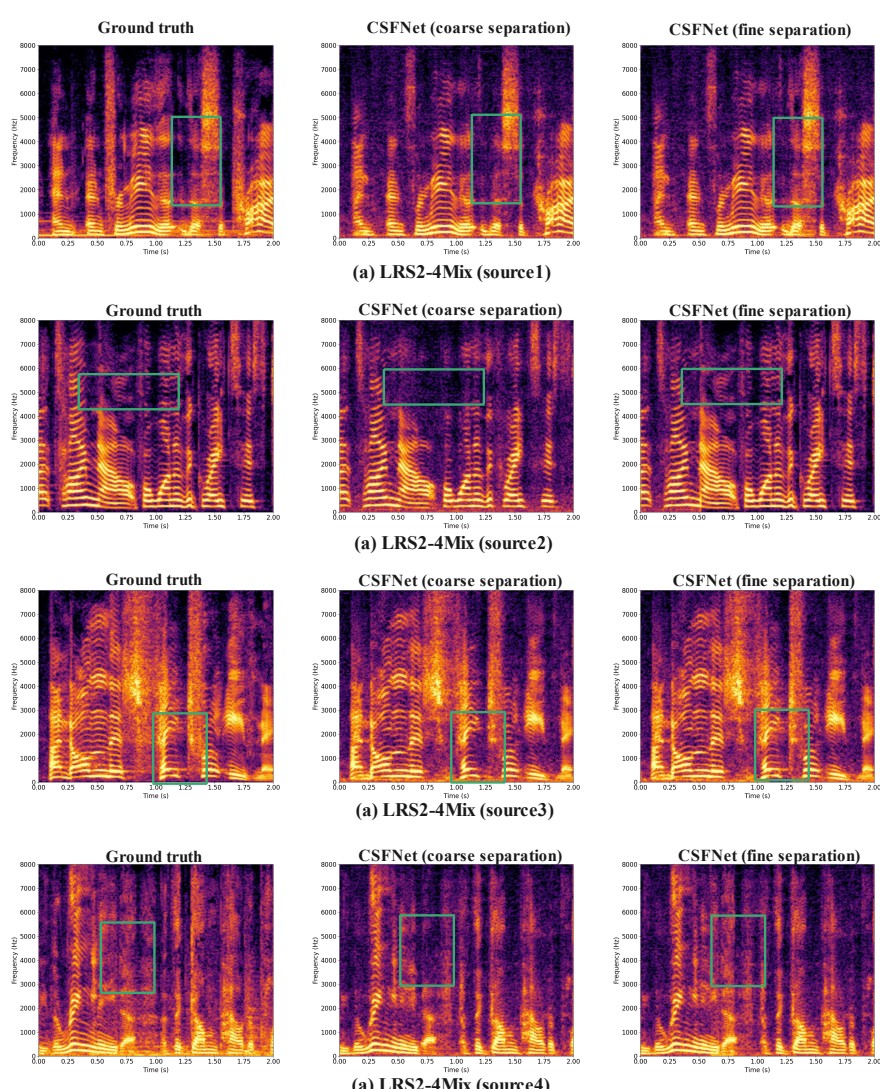

Figure 9: Comparison of the spectrograms of the ground truth, audio separated by CSFNet (coarse separation) and by CSFNet (fine separation).

