# OpenReview forum: "From Coarse to Fine: Recursive Audio-Visual Semantic Enhancement for Speech Separation"
_ICLR.cc/2026/Conference — ICLR 2026 Conference Withdrawn Submission_

### Official Review · Reviewer_WpMN · 2025-10-29

**Soundness:** 3
**Presentation:** 3
**Contribution:** 3
**Rating:** 6
**Confidence:** 5

**Summary:**

This paper proposes CSFNet, a coarse-to-fine framework for audio-visual speech separation that recursively enhances semantic representations. The method first performs coarse separation to obtain preliminary speech estimates, which are then fed into a pretrained Auto-AVSR model along with lip features to derive refined audio-visual semantics for fine separation. Additionally, the authors design a Speaker-wise Perceptual (SP) fusion block for robust cross-modal fusion and a Multi-range Spectro-Temporal (MST) separation module to capture both local and global time-frequency dependencies. Extensive experiments on multiple datasets (LRS2, LRS3, VoxCeleb2, NTCD-TIMIT, LRS3+WHAM!) demonstrate consistent state-of-the-art performance and robustness in noisy or occluded-visual conditions.

**Strengths:**

1.	Novel recursive semantic enhancement strategy. The idea of feeding back coarse audio into an AVSR model for semantic refinement is conceptually appealing and empirically effective. It provides a meaningful link between speech recognition and separation.
2.	Strong performance and comprehensive experiments. The proposed method achieves clear SOTA results across multiple benchmarks, including both clean and noisy datasets, with consistent gains in SI-SDRi, PESQ, and WER metrics. Notably, the second stage of the proposed two-stage framework yields substantial performance gains in more challenging multi-speaker scenarios, particularly under 3-mixture and 4-mixture conditions.
3.	Well-structured ablations. The authors thoroughly analyze the contribution of each component (two-stage design, SP fusion, MST module, visual occlusion tests), which strengthens the credibility of the method.
4.	Clear presentation and solid engineering. The paper is well written, figures are clear, and the methodology is reproducible. The design choices are motivated and technically sound.

**Weaknesses:**

1.	Clarification on visual frame occlusion simulation (Figure 4 & Appendix F). Figure 4 presents results for cases where one or both speakers have partially missing visual frames. As described in Appendix F, the authors simulate missing 5, 10, 20, 30, 40, and all frames. However, for the case where both speakers lose visual cues, it is not clearly stated whether the missing frame segments are temporally aligned or occur at different timestamps. In practice, manually discarding frames may cause temporal misalignment between audio and video streams, which could significantly influence model performance. The lack of detailed description on how this issue is handled makes it difficult for readers to fully assess the validity of the visual occlusion experiments.
2.	The paper introduces a simple yet efficient fusion module, the Speaker-wise Perceptual (SP) fusion block. However, the authors do not explain why chose not to adopt common attention-based fusion mechanisms that are widely used in previous audio-visual separation works.
3.	Inference efficiency and online applicability. Since the proposed system is composed of two sequential stages (coarse and fine separation), it appears that inference also requires two passes through the network: the first to generate coarse audio and the second to refine it. This could potentially double the inference time, making the method less suitable for real-time or low-latency applications.
4.	Table 5 suggests that both stages rely heavily on the pretrained Auto-AVSR model. This raises a concern that the performance improvements and SOTA results might largely stem from the strength of the pretrained AVSR backbone, rather than the proposed recursive separation architecture itself. A more thorough discussion on this trade-off would strengthen the paper.

**Questions:**

Consistent with Weakness.

---

> ### Author Response · Authors · 2025-11-19
>
> **To weakness1:**  Thank for your thoughtful observation. We agree that frame-level temporal alignment between audio and video streams is crucial for audio-visual speech separation performance. We appreciate the opportunity to clarify the experimental setup of the visual occlusion simulation for better transparency and reproducibility.
> We replace Appendix F as follows:
>
> We follow prior works and process the video input at 25 frames per second over a 2-second window, resulting in 50 frames per sample. To simulate partial visual occlusion in the two-speaker mixed data, we consider two scenarios: (1) Only one speaker's visual input is partially occluded. We randomly remove a consecutive block of 5, 10, 20, 30, 40, or all frames. For example, in the case of 10 missing frames, we randomly choose a starting position and replace the subsequent 10 frames with zero-valued frames. This operation ensures that the temporal structure and frame alignment of the sequence are preserved despite the simulated occlusion. (2) Both speakers' visual inputs are partially occluded. Each speaker has the same number of missing frames overall, but the occluded segments are applied independently and do not occur at the same timestamps. For instance, for 10 missing frames, each speaker loses a random consecutive block of 10 frames at different positions. As in the first scenario, all removed frames are replaced by zero-valued frames so that the overall temporal continuity and alignment of the video streams remain intact.
>
> **To weakness2:** Exactly, the design of the Speaker-wise Perceptual (SP) fusion block is a key contribution of our work. We chose not to adopt commonly used attention-based fusion mechanisms for the following reasons:
> 1. The SP fusion block is designed to explicitly capture both global and local relationships between each speaker’s visual features and the mixture speech representation. Unlike traditional attention-based fusion, which typically models cross-modal interactions more generally, our SP block incorporates speaker-wise conditioning, making it more suitable for multi-speaker scenarios where each speaker's characteristics must be modeled independently.
> 2. As shown in Figure 5, SP fusion consistently outperforms attention-based fusion methods when visual information is partially missing. This demonstrates that SP fusion better handles visual uncertainty by maintaining stable alignment and incorporating audio-driven cues, which is essential in realistic conditions where visual occlusion is common.
>
> **To weakness3:** It is true that our framework performs two sequential inference passes: the first stage produces a coarse audio estimate, and the second stage refines it. This naturally introduces additional inference latency compared to single-stage systems.
> However, we would like to clarify two key aspects:
>
> 1. Our core motivation is to enhance downstream ASR performance by improving separation accuracy, particularly under challenging multi-speaker and noisy conditions. For such scenarios, sacrificing a degree of latency in exchange for substantial quality gains is often acceptable. Nevertheless, our framework is designed to be flexible: the coarse stage itself is fully functional and achieves SOTA performance on its own, as shown in Tables 1-4. By omitting the refinement stage, the system can operate in real-time or low-latency environments while still providing competitive performance.
>
> 2. The two-stage design allows practical deployment strategies. For instance, the coarse stage can be used for real-time streaming, and the fine stage can be applied offline for high-accuracy back-end transcription or archival purposes. This separation of functionality offers greater adaptability across application scenarios, without requiring architectural changes.
>
>
> **To weakness4:** We appreciate the reviewer’s thoughtful observation. First, as shown in Table 1, even without incorporating visual information, **CSFNet (Audio-Only)** achieves SOTA performance in the two-speaker speech separation task, outperforming prior audio-only models. This demonstrates that the architecture itself contributes significantly to the separation quality, independent of visual or pretrained components. The multi-scale temporal processing and recursive separation mechanism play a vital role in this improvement. Second, Table5 further clarifies the role of the pretrained AVSR model. While Auto-AVSR indeed provides stronger initial representations compared to Deep-AVSR or AV-HuBERT, all pretrained models benefit consistently from the second (fine) stage. Specifically, the Word Error Rate (WER) reduces from 24.94 to 21.78 for Deep-AVSR, 24.66 to 21.50 for AV-HuBERT, and 22.75 to 18.89 for Auto-AVSR. These improvements across the board validate the effectiveness of the second-stage refinement, independent of the specific pretrained model used.

---

> > ### Comment · Reviewer_WpMN · 2025-11-26
> >
> > Thank you for the response. The rebuttal addressed some of my concerns. I am raising the scoring from 6 to 8.

---

> > > ### Author Response · Authors · 2025-11-27
> > >
> > > Dear Reviewer WpMN,
> > >
> > > Thank you for your positive feedback and for raising your score. We are very glad to hear that our rebuttal successfully addressed your concerns. Your insightful comments during the review process have been invaluable in strengthening our paper.
> > >
> > > Best regards,
> > >
> > > Authors.

---

### Official Review · Reviewer_gVy4 · 2025-10-29

**Soundness:** 2
**Presentation:** 4
**Contribution:** 2
**Rating:** 4
**Confidence:** 4

**Summary:**

This paper proposes CSFNet (Coarse-to-Separate-Fine Network), a two-stage audio-visual speech separation system that introduces a recursive semantic enhancement paradigm. The approach operates in two stages: (1) a coarse separation stage that produces an initial audio estimate from mixed audio and visual input, and (2) a fine separation stage that feeds the coarse audio back into an audio-visual speech recognition (AVSR) model together with visual information to extract enhanced semantic representations for refined separation. The model incorporates a speaker-aware perceptual (SP) fusion block and a multi-range spectro-temporal (MST) separation module. Extensive experiments on three benchmark datasets (LRS2, LRS3, VoxCeleb2) and two noisy datasets demonstrate state-of-the-art performance with substantial coarse-to-fine improvements.

Overall, the other experimentation in the paper and the Appendix is solid and detailed, a very interesting read! The model has very strong performance, and outperforms competitors by a wide margin in a wide variety of tasks. However, the model is similar to previous methods in many ways, and so the contributions in terms of model architecture are too minimal to meet the standards of ICLR. That being said, given the strong performance and detailed results, I would be willing to accept the paper if the issues detailed in this review are addressed with further experimentation and justification for the choices made.

**Strengths:**

The model demonstrates very strong performance, outperforming competitors by a wide margin across a wide variety of tasks and datasets, including both clean and noisy conditions. The extensive experimentation presented in both the main paper and appendix is solid and detailed, providing a very interesting and thorough evaluation. The recursive coarse-to-fine framework presents an interesting approach to leveraging audio-visual semantic information more effectively than prior work that relies solely on visual input for semantic extraction.

**Weaknesses:**

Speaker-wise Perceptual (SP) Fusion Block is fairly similar to the fusion approach in CTCNet, with a simple adaptation to the TF domain. The exact same adaption to the TF domain was also used as a baseline in the RTFS-Net paper, which similarly interpolates in the time dim, and then broadcasts across the frequency dim (see RTFS-Net section 5.2). This mixture of approaches is fine, but is very similar to the work covered by these two papers and hence should be cited. In addition, when evaluating performance, SP fusion should compare to the previous baselines, such as the fusion methods in CTCNet, RTFS-Net, AVLiT or the current SOTA methods AVSepChain and IIANet. In addition, Figure 5 is quite difficult to read. The colours are very similar. A black boarder around each bar would aid visibility. A table would be even better. Also, RTFS-Net's results are also missing from table 1.

The order of the results in the tables is a little confusing. It's not by year or performance. To help the reader, please put the models in the table in order of performance with the best result highlighted in bold. For example, in table 2 you highlight your method, but the best method for macs, params, time and memory are not your method. This is fine, but please clear this up to make things clear.

The main focus of the paper is a coarse to fine refinement process - running the audio through the model twice. However, it is not clear from the results where the strong performance of the paper is coming from. The current SOTA AVSS method is IIANet, which uses a different pre-trained video encoder. It's hard to see if CSFNet's performance comes from the new video encoder i.e. pre-trained Auto-AVSR (Ma et al., 2023), or the actual model architecture. For example, if you swapped CSFNet's video encoder with IIANet's video encoder, or conversely, replace IIANet's video encoder with pre-trained Auto-AVSR (Ma et al., 2023) and ran IIANet twice in a course-fine manner, which approach would have the better performance?

In table 2, it is unclear if these results are for the course+fine CSFNet or the course only approach. Therefore, it could be interpreted that the coarse-only method only barely outperforms IIANet, is slower, and takes up significantly more memory. Please clarify! It would also be interesting to see the course only model macs and parameters.

**Questions:**

The encoder is interesting, using multi-scale dilations to gather larger and larger context. Most other methods in AVSS use a simple 1D or 2D convolution depending on if the model is working with the time domain or the TF domain, would it be possible to see some experimentation on the benefits of this encoder over the encoder used in other methods? There needs to be some justification for this added complexity + parameters.

The main (MST) separation module is similar to TF-GridNet, which was also used by RTFS-Net for AVSS. As with RTFS-Net, you make some adaptations, including a triple unfold of the features and some kind of fusion. However, the paper does not detail how this fusion works - summation/concatenation/attn/projection? You only state "The outputs of these branches are fused to produce richer multi-scale representations...". Papers need to be repeatable, and this mechanism is unclear. The ablations studies in Appendix C were well done and detailed though. Very interesting results + good efficiency gains with the tweaks to BLSTM.

In table 1, there is an audio only model, but it is unclear how this audio only model works. Is it just CSFNet without the SP fusion block and no video inputs?

The course to fine approach is interesting, but can it be extended? What if we run the audio through the model 3 times, do we continue to get a higher quality separation? Or do we reach a saturation point?

---

> ### Author Response · Authors · 2025-11-19
>
> **To weakness1:** We have cited the mentioned two papers in the revised manuscript. To further validate the efficiency
> of our proposed SP fusion block, we replaced it with the fusion module (termed CAF Block) from
> RTFS-Net in the first stage and conducted comparative experiments. We maintained our model’s
> configuration parameters, where the speech channel count is 192 and the video channel count is 768.
> The experimental results are summarized in the below table:
>
> | Method                  | SI-SDRi (↑) | Params (M) | MACs (G) |
> |-------------------------|-------------|------------|-----------|
> | SP Fusion Block          | 16.2        | 0.1        | 0.3       |
> | CAF Block (RTFS-Net)     | 16.4        | 1.8        | 21.56     |
> |
>
>
>
> It can be observed that adopting the CAF fusion method improves performance, but under our configuration, its parameter count increases by an order of magnitude. In particular, the computational
> complexity (MACs) increases significantly (from 0.3G to 21.56G). Therefore, using CAF is not very
> reasonable in our scenario.
>
> Regarding the fusion methods of CTCNet and AVLiT, we have not yet run the corresponding experiments due to time constraints, and we will continue to conduct them on the go for comparison.
> Briefly, CTCNet performs dynamic multi-scale audio-visual fusion through a cyclic architecture with
> hierarchical unimodal cross-layer interactions, while AVLiT fuses modalities by simply adding audio
> and video features. Both approaches differ from our fusion design.
>
> Besides, we have revised Figure 5 by enhancing the contrast between the bars and adding a clear black
> border around each bar to improve visibility. If the reviewer still finds the updated figure difficult to
> interpret, we can also replace the figure with a table of numerical results in the final version of the
> paper. Besides, RTFS-Net has been added into Table 1.
>
>
> **To weakness2:** To improve readability and facilitate fair comparison, we have revised all tables to sort the models in
> descending order of performance. Additionally, for each metric, the best result is now consistently
> highlighted in bold, regardless of whether it is achieved by our method or a baseline. This ensures
> that readers can easily identify both overall trends and specific strengths of each method without
> confusion. Thank you again for the constructive feedback.
>
>
> **To weakness3:** Thank you for raising this insightful question. To clearly disentangle the contributions of the proposed
> architecture from those of the pretrained video encoder, we conducted additional controlled experiments. Specifically, we compared (1) our CSFNet and IIANet under the audio-only condition, and
> (2) CSFNet equipped with IIANet’s video encoder (CTCNet-Lip). The results are summarized in the
> following table.
>
> | Method               | LRS2 SI-SDRi ↑ | LRS2 SDRi ↑ | LRS3 SI-SDRi ↑ | LRS3 SDRi ↑ | VoxCeleb2 SI-SDRi ↑ | VoxCeleb2 SDRi ↑ |
> |----------------------|----------------|-------------|----------------|-------------|--------------------|-----------------|
> | IIANet (Audio-Only)  | 11.2           | 11.5        | 12.5           | 12.8        | 9.5                | 10.0            |
> | CSFNet (Audio-Only)  | 14.5           | 14.6        | 17.4           | 17.7        | 11.9               | 12.5            |
> | CSFNet (CTCNet-Lip)  | 16.0           | 16.2        | 18.5           | 18.6        | 13.9               | 14.3            |
> | CSFNet (Auto-AVSR)   | 16.2           | 16.3        | 18.6           | 18.8        | 14.1               | 14.4            |
> |
>
>
> 1. Evidence from the Audio-Only Setting. When no visual information is used, CSFNet(Audio-Only) which retains only the coarse stage, significantly outperforms IIANet(Audio-Only). This demonstrates that the performance gain does not merely come from the pretrained Auto-AVSR encoder, but
> strongly reflects the advantage of our model architecture itself.
> 2. Swapping Video Encoders. When replacing Auto-AVSR with the CTCNet-Lip encoder used by
> IIANet, the performance of CSFNet decreases slightly but remains highly competitive. This indicates that: 1. The architecture of CSFNet is robust and not overly dependent on a particular video encoder, 2. Auto-AVSR further enhances the performance, but is not the root cause of CSFNet’s improvements.

---

> > ### Author Response · Authors · 2025-11-19
> >
> > **To weakness4:** Thank you very much for pointing this out. We apologize for the confusion caused. We have now
> > updated Table 2 to clearly distinguish between the coarse-only and coarse+fine (two-stage) versions
> > of CSFNet. This provides a more accurate reflection of the performance and resource demands of
> > each stage of CSFNet.
> > - As shown in Table 2, when only the coarse separation stage is used, our model already outperforms IIANet (which is further reflected under noisy conditions (Table 3) and in the scenarios
> > with more than two speakers (Table 4)), while requiring only about 1.5x MACs and exhibiting
> > nearly identical GPU inference time. This trade-off is acceptable given the performance gains,
> > and places our model within the same computational range as IIANet.
> > - For the turth that our model contains roughly 3× more parameters than IIANet, this is primarily because IIANet employs parameter-sharing across blocks, which significantly reduces both
> > parameter count and memory usage. In contrast, our model maintains independently parameterized blocks for better representational capacity. It is worthy noting that, compared to other
> > SOTA models, such as AV-Crossnet, our parameter count and MACs remains within the
> > same order of magnitude, indicating that the model is not excessively large in a broader context.
> > - In the fine separation stage, we perform a lightweight fine-tuning of the coarse model, reusing
> > the same parameters via sharing. Therefore, the total number of trainable parameters remains
> > unchanged. The additional computational cost comes from processing intermediate representations twice, which doubles the MACs but not the model size.
> >
> >
> > **To question1:** We sincerely appreciate your constructive suggestion, and we are very happy to conduct the additional
> > experiments you proposed to further investigate the benefits of our multi-scale dilated encoder. In
> > response, we carried out a detailed comparison between our encoder and a standard convolutional
> > encoder (using a fixed 3×3 kernel) under the same feature dimension.In our encoder, the feature dimension is evenly split into four branches, each using convolutional
> > kernels of sizes 1×1, 3×3, 5×5, and 7×7 with increasing dilation rates. This design effectively
> > enlarges the receptive field without introducing additional computational burden. In fact, due to the
> > use of dilated convolutions, our encoder requires fewer parameters and MACs than the standard
> > convolutional encoder:
> >
> > - Parameters: 2880 (ours) vs. 3648 (standard)
> > - MACs: 0.09G (ours) vs. 0.10G (standard)
> >
> > Furthermore, this multi-scale receptive field leads to measurable performance improvements, yielding
> > an SI-SDR gain of **+0.05 to +0.1 dB** over the standard encoder. These results confirm the effectiveness
> > of our encoder in delivering a better trade-off between performance and architectural simplicity.
> >
> >
> > **To question2:**
> > We sincerely thank the reviewer for highlighting the need for clarity regarding the fusion mecha-
> > nism in the MST separation module. We apologize for the lack of explicit detail in the initial submission. To address this, we have revised the manuscript to include a more detailed description of
> > the fusion strategy on Page 5 (3.2.4) of the revised manuscript as follows: The outputs from all
> > branches are concatenated along the channel dimension and subsequently processed by a 2D convolution layer, producing richer and more informative multi-scale representations, improving performance
> > in overlapping-speaker scenarios. We have also added new Figure 7 in Appendix C to visualize the
> > architecture of the MST module and clarify the fusion mechanism in a reproducible manner.
> > We further appreciate the reviewer’s positive feedback on our ablation studies in Appendix C. We are
> > glad to hear that the observations on efficiency improvements and the BLSTM modifications were
> > found interesting and well documented.

---

> ### Author Response · Authors · 2025-11-19
>
> **To question3:**
> Thank you for your helpful comment and for pointing out this oversight. We apologize for not making
> the details of the audio-only model sufficiently clear in the original manuscript. As correctly inferred
> by the reviewer, the CSFNet (audio-only) model is implemented by simply removing the SP fusion
> block and the corresponding visual input, such that the model goes through only the coarse stage; all
> other components and settings of CSFNet remain unchanged. We have clarified this in implementation
> details of the revised version of the paper to avoid any ambiguity.
>
> **To question4:**
> Thank you for this insightful question. The idea of iterating the coarse to fine refinement multiple
> times is indeed interesting and technically feasible. To investigate this, we conduct additional experiments on the LRS2-2Mix dataset by extending our model to a third and fourth iteration, and present
> the results below.
>
> | Method                     | SI-SDRi | MACs (G) | Params (M) |
> |----------------------------|---------|-----------|------------|
> | CSFNet (coarse separation) | 16.2    | 31.7      | 10.9       |
> | CSFNet (fine separation)   | 16.8    | 64.8      | 10.9       |
> | CSFNet (3rd iter)          | 17.1    | 97.1      | 10.9       |
> | CSFNet (4th iter)          | 17.2    | 121.2     | 10.9       |
> |
>
>
> The results show that the second iteration, namely the fine separation stage, provides the
> most substantial performance improvement. In contrast, the third and fourth iterations yield only
> marginal gains while significantly increasing computational cost, memory usage, and inference time.
> A saturation of performance can be expected as the iteration round further grows. Therefore, one
> refinement step offers the a really good balance between performance and efficiency.

---

> ### Author Response · Authors · 2025-11-22
>
> **To weakness 1**: On the fusion methods of CTCNet and AVLiT, we have completed experimental validation and conducted a systematic comparison with CAF and the SP fusion module, as shown in the table below. The analysis indicates that while AVLiT employs a simple modality summation approach, resulting in the smallest parameter count and lowest computational complexity, its performance is also the least satisfactory. In contrast, CTCNet's fusion method achieves performance comparable to the CAF block but with both fewer parameters and lower computational complexity. When compared to our proposed SP fusion method, CTCNet achieves a performance improvement of 0.2 dB; however, this comes at the cost of approximately 5 times more parameters and 8 times higher computational complexity.
>
>
> | Method                  | SI-SDRi (↑) | Params (M) | MACs (G) |
> |-------------------------|-------------|------------|-----------|
> | Summation (AVLiT)         | 16.0        | 0.1        | 0.2       |
> | CTC-Fusion (CTCNet)     | 16.4        | 0.5        | 2.4     |
> | CAF Block (RTFS-Net)     | 16.4        | 1.8        | 21.56     |
> | **SP Fusion Block (Ours)**          | 16.2        | 0.1        | 0.3       |
> |

---

> > ### Author Response · Authors · 2025-11-28
> > **Reason for the Significant Performance Improvements Achieved by Our Model**
> >
> > Dear Reviewer,
> >
> > We would like to provide further clarification on the key innovations of our work, as they are also the reasons behind its significant performance improvements:
> >
> > Currently, existing audio-visual speech separation models primarily pursue performance gains from three perspectives: 1) Optimizing the model architecture itself (e.g., AV-Crossnet, AV-ConvTasNet); 2) Leveraging powerful pretrained models (e.g., AV-HuBERT) to extract more accurate semantic information from video for auxiliary separation (e.g., AV-SepChain); 3) Developing more efficient audio-visual fusion methods (e.g., CTCNet). However, regarding the second perspective, even state-of-the-art pretrained video feature extractors have inherent limitations in capturing precise semantic information within short durations (e.g., 2 seconds), leading to a performance bottleneck.
> >
> > Our work is the first in the field to combine audio and video to capture more precise and refined semantic features, rather than merely utilizing video alone. Compared with video-only inputs, this approach significantly improves the accuracy of semantic feature extraction, which in turn contributes to substantial performance gains. Additionally, it enhances robustness when parts of the video are missing. From this perspective, our method proposes a new fusion paradigm that sheds light on how to more effectively process and utilize visual features while fully leveraging audio information. We believe this can constitute a fourth perspective for advancing audio-visual speech separation models, warranting further research in the future.
> >
> > We greatly value the opportunity to communicate with you further. With your guidance, we are confident that the quality of our paper will be significantly enhanced. Please do not hesitate to contact us with any additional questions or concerns, and we will make every effort to address them promptly.
> >
> > Thank you very much for your time and consideration. We wish you all the best.
> >
> > Sincerely,
> > The Authors

---

### Official Review · Reviewer_28g8 · 2025-10-31

**Soundness:** 3
**Presentation:** 3
**Contribution:** 3
**Rating:** 2
**Confidence:** 4

**Summary:**

This paper proposes the Coarse-to-Separate-Fine Network (CSFNet) to improve audio-visual speech separation (AVSS) performance. Unlike existing AVSS models that rely on fixed visual information, CSFNet recursively enhances audio-visual representations through a two-stage structure consisting of coarse and fine separation. Specifically, in the first stage, a coarse speech separation is performed, and its output is reused as input to an audio-visual speech recognition (AVSR) model to generate more discriminative semantic representations, which are then leveraged to perform the fine separation. In addition, we introduce the Speaker-wise Perceptual (SP) Fusion Block and the Multi-range Spectro-Temporal (MST) Separation Module to accurately capture speaker-specific cues and temporal information. Experimental results demonstrate that CSFNet achieves higher separation performance than existing SOTA methods while exhibiting strong robustness under noisy conditions.

**Strengths:**

* Through recursive semantic enhancement, the coarse separation output is fed back into the AVSR model to learn stronger audio-visual semantic representations—a concept that is novel and empirically leads to performance improvements.

* Experimental results show that performance improves noticeably from the coarse to the fine stage, and it is particularly noteworthy that the model achieves strong performance in multi-speaker scenarios and when visual cues are missing.

**Weaknesses:**

* Tables 1 and 2 show that the performance improvement does not scale proportionally with the increase in parameters. In particular, compared to IIANet, CSFNet has roughly three times more parameters, but the performance gain is quite marginal. While the authors claim a substantial improvement in SI-SDR over IIANet, considering that CSFNet goes through both coarse and fine separation stages, the performance improvement seems insufficient.
* There is a lack of analysis on the specific roles of the coarse and fine stages. In extreme cases, it is difficult to distinguish the effect from simply training for twice as many steps. A more detailed analysis, for example using attention maps to show the distinct contributions of the coarse and fine stages, would strengthen the study.
* Compared to prior work, the approach gives the impression of simply adding an extra stage, which makes the novelty aspect somewhat limited.

**Questions:**

* Would it be possible to provide a comparison with IIANet and AVSepChain under noisy conditions in Table 3 as well?
* Although the authors refer to the method as coarse and fine separation, it seems that it could in fact be applied iteratively for multiple repetitions. Could you provide experimental results showing how performance improves when each stage is repeated multiple times?
* Since the dual-stage structure trains and infers both the coarse and fine stages, it appears that the inference time and parameter count would roughly double. Could you clarify whether the parameter values reported in Tables 1 and 2 account for this?

---

> ### Author Response · Authors · 2025-11-19
>
> **To weakness1:** We appreciate the reviewer’s insightful observation. We would like to clarify that Tables 1 and 2 report
> results on a clean two-speaker dataset, where the experimental setting is relatively simple. Under these
> conditions, the performance difference between our model and IIANet is indeed moderate. However,
> our method is designed for more challenging and realistic scenarios, which is where its advantages
> become evident:
> - As shown in Table 3, our model achieves significantly higher SI-SDR under noisy conditions
> compared to IIANet, demonstrating superior robustness when confronted with real-world noise,
> an area where audio-visual fusion is particularly beneficial.
> - In more complex scenarios involving more than two speakers (Table 4), our model significantly
> outperforms IIANet across all configurations, indicating stronger scalability and generalization.
> This is especially important for real-world applications such as meeting transcription or cocktail party environments, where speaker count varies and mixtures are inherently complex.
> - Although our model contains approximately three times more parameters than IIANet, this is
> largely due to the absence of parameter-sharing across blocks, a deliberate choice to preserve
> feature diversity, as adopted by IIANet. When it comes to inference latency, the advantage
> achieved by parameter-sharing structure degrades to be around twice at most. Notably, as shown
> in Table 2, the computational complexity (MACs) and inference latency of our coarse stage
> remain comparable to those of IIANet, while still achieving superior SI-SDR performance.
> - Compared to other SOTA methods, with our model size (around 10M parameters) remains
> within a normal range. Besides CSFNet provides consistently higher performance across both
> standard and challenging conditions, highlighting its efficiency and practicality.
>
>
> **To weakness2:** Thank you for this insightful question. In audio-visual speech separation, the visual modality plays a
> crucial role: the more accurate the extracted visual features are, the better the separation performance
> becomes. When the pretrained Auto-AVSR model receives only video as input, its recognition accuracy is relatively limited (around 70%, as shown in Table 5), meaning that the visual cues it provides
> for speech separation are not highly reliable.
>
> The purpose of the coarse stage is therefore to generate an initial speech estimate that is sufficiently
> intelligible for Auto-AVSR. With this improved audio–visual input pair, Auto-AVSR produces noticeably more accurate visual representations, improving its WER by roughly 10% (from about 70%
> to nearly 80%). These more reliable visual features are then fed into the fine stage, whose goal is to
> leverage the strengthened cross-modal alignment to extract a more refined speech signal.
>
> Similar to our discussion in the response to Reviewer gVy4 Question 4, each refinement step further
> improves the separation quality but with diminishing returns. After the second stage, the speech quality becomes sufficiently high such that the additional improvement in visual representations becomes
> marginal, leading the refinement process to gradually saturate. This demonstrates that the coarse and
> fine stages play distinct roles: the coarse stage enhances audio–visual alignment by improving intelligibility, while the fine stage exploits the more accurate visual cues to refine the separation output.
>
> **To weakness3:** We would like to clarify that, in addition to the coarse-to-fine structure, we have introduced several
> new designs. As shown in Tables 1, 3, and 4 of the paper, the coarse-stage model (CSFNet coarse)
> alone achieves strong performance, outperforming existing methods. This success can be attributed to
> the innovative architectural design of the CSFNet in the coarse stage. The key features of this design
> include:
> - As described in Section 3.2.4, the MST module leverages multiple unfold operations to capture
> a broader context. Compared to the original TF-GridNet module, this design not only expands
> the receptive field but also significantly reduces both the parameter count and computational
> complexity, while improving separation quality.
> - As detailed in our rebuttal to Question 1 of Reviewer gVy4, we introduce a multi-range dilated
> convolutional encoder. By splitting the channel into four parallel branches (with kernel sizes
> 1 × 1, 3 × 3, 5 × 5, and 7 × 7), each using dilated convolutions, the encoder efficiently captures
> multi-scale features. This design further reduces parameter size and computational demands,
> while providing measurable gains in model performance.

---

> > ### Author Response · Authors · 2025-11-19
> >
> > **To question1:** We thank the reviewer for this helpful suggestion. Following your request, we have updated Table 3 to
> > include the performance of IIANet under two investigated noisy conditions. Unfortunately, we were
> > unable to include AVSepChain in our comparisons as its implementation is not publicly available, and
> > no pretrained model or code is provided in their publication.
> >
> > As shown in the revised Table 3, our proposed CSFNet, especially when using both the coarse and fine
> > separation stages, consistently outperforms IIANet under all the noise settings. This further validates
> > the robustness and effectiveness of our approach in realistic noisy environments.
> >
> >
> >
> > **To question2:** Thank you for your insightful question. Indeed, our proposed two-stage (coarse-to-fine) separation
> > framework can be naturally extended to support iterative refinement, which is the same as the Question
> > 4 of reviewer gVy4: Each iteration refines the initial separation by progressively enhancing the target
> > speech and suppressing interference, making the architecture suitable for multi-stage stacking.
> >
> > We have conducted additional experiments to evaluate the performance when the separation process is
> > applied repeatedly. Specifically, we tested configurations with 1, 2 and 3, 4 iterations of the two-stage
> > separation pipeline. The results, presented in the following table, demonstrate consistent performance
> > gains with iterative refinement, though with diminishing returns beyond 2 iterations. These findings
> > indicate that while iterative application enhances performance, a small number of iterations (typically
> > two) already achieves a good balance between performance and computational cost.
> >
> > | Method| SI-SDRi | MACs (G) | Params (M) |
> > |-|-|-|-|
> > | CSFNet (coarse separation) |16.2 | 31.7     | 10.9       |
> > | CSFNet (fine separation)   | 16.8 | 64.8     | 10.9       |
> > | CSFNet (3rd iter)          | 17.1 | 97.1     | 10.9       |
> > | CSFNet (4th iter)          | 17.2 | 121.2    | 10.9       |
> > |
> >
> >
> > **To question3:** In Table 2, we have now separated the results of the two stages of our model for clearer comparison.
> > Regarding the parameter count in Table 1 and updated Table 2, both the CSFNet (coarse separation)
> > stage and the CSFNet (fine separation) stage contain 10.9M parameters. The parameter count in two
> > stages does not double because the second stage is initialized from the first stage’s output and fully
> > shares its weights, keeping the total number of parameters at 10.9M.

---

> > > ### Author Response · Authors · 2025-11-28
> > >
> > > Dear Reviewer,
> > >
> > > Regarding Weakness 3 concerning the key novelty of our work, we would like to provide further clarification:
> > >
> > > While the coarse-to-fine workflow itself，i.e., adding an additional stage，is a classic and widely used paradigm, the underlying perspective we adopt in the field of audio-visual speech separation is highly innovative. The detailed reasoning is as follows:
> > >
> > > Currently, existing audio-visual speech separation models primarily pursue performance improvements from three perspectives: 1) Optimizing the model architecture itself (e.g., AV-Crossnet, AV-ConvTasNet); 2) Leveraging powerful pretrained models (e.g., AV-HuBERT) to extract more accurate semantic information from video for auxiliary separation (e.g., AV-SepChain); 3) Developing more efficient audio-visual fusion methods (e.g., CTCNet). However, for the second perspective, even state-of-the-art pretrained video feature extractors have inherent limitations in capturing accurate semantic information within short durations (e.g., 2 seconds), resulting in a performance bottleneck.
> > >
> > > Our work is the first in the field to combine audio and video to capture more precise and refined semantic features, rather than extracting them solely from video. Compared with video-only inputs, this approach significantly improves the accuracy of semantic feature extraction, which in turn drives substantial performance gains. From this perspective, our method proposes a new fusion paradigm that sheds light on how to more effectively process and utilize visual features while fully leveraging audio information. We believe this can constitute a fourth perspective for advancing audio-visual speech separation models, warranting further research，for instance, aligning semantic information in visual inputs with the aid of speech, without relying on powerful yet parameter-heavy AVSR models.
> > >
> > > We greatly value the opportunity to communicate with you further. With your guidance, we are confident that the quality of our paper will be significantly enhanced. Please do not hesitate to contact us with any additional questions or concerns, and we will make every effort to address them promptly.
> > >
> > > Thank you very much for your time and consideration. We wish you all the best.
> > >
> > > Sincerely,
> > > The Authors

---

### Official Review · Reviewer_Eb18 · 2025-11-04

**Soundness:** 3
**Presentation:** 3
**Contribution:** 2
**Rating:** 6
**Confidence:** 4

**Summary:**

The paper presents a new method for audio-visual speech separation. The main idea is to separate the the mixed audio twice. In the first stage, it produces coarse results. In the second stage, it separates the original mixed audio again, but this time, it is aided by the results obtained in the first stage. The coarse results are integrated into the video encoder, which produces better fused audio-visual features for the model to do the fine separation in the second stage. The results show a good trade-off between the accuracy and computational cost.

**Strengths:**

1. The main idea -- from coarse to fine -- is simple and makes sense.
2. The results are extensive, and show a good tradeoff between the quality of the results and the computational cost. In addition, in the conditions of more than 2 speakers (Table 4), the proposed method outperformed other methods by a large margin.
3. Presentation is good. It's easy to understand the contents.

**Weaknesses:**

1. The idea is straightforward, not novel enough, considering that the principle of coarse-to-fine has been used in model design in many areas including computer vision and audio processing. In fact, the iterative nature of A-FRCNN  (Hu et al., 2021) and CTCNet (Li et al., 2024a) also makes them follow into the coarse-to-fine model category.
2. The computational cost is high. The reason is clear: the model needs to process the audio signal and video signal twice. From Table 2, the size of the model and the memory usage are more than 3X of a state-of-the-art model IIANet. This could be partially sensed by the expensive computing facility used in experiments:  8 NVIDIA A100 GPUs (40 GB).

**Questions:**

In the conditions of more than 2 speakers (Table 4), the proposed method outperformed other methods by a large margin. Besides the coarse-to-fine strategy, is there any other reason?

In the SP fusion block (Figure 3a), the features of the audio mixture are concatenated with two speakers' video features. Does it mean that we need to train a model for each n, where n is the number of speakers? Please note that in many previous audio-visual speech separation models, such as (Pegg et al. 2024), this is not necessary. Is this a reason of good results for more than 2 speakers?

---

> ### Author Response · Authors · 2025-11-19
>
> **To weakness1:** Besides the coarse-to-fine structure, our model introduces novel architectural designs:
>
> Unlike prior works such as A-FRCNN (Hu et al., 2021) and CTCNet (Li et al., 2024a), which iterate the same network block 4–12 times, our model performs only two iterations and saturates within four rounds (as discussed in our rebuttal to Question 4 of Reviewer gVy4). This shows that the coarse separation stage is fundamentally different. Moreover, Tables 1, 3, and 4 demonstrate that CSFNet (coarse) alone already achieves strong performance. Specifically:
>
> 1. MST module: Uses multiple **unfold** operations to expand context. Compared with TF-GridNet, it increases the receptive field while reducing parameters and computational cost, improving separation quality.
>
> 2. Multi-range dilated convolutional encoder: Channels are split into four branches (kernels $1\times1$, $3\times3$, $5\times5$, $7\times7$) with dilated convolutions, capturing multi-scale features efficiently, reducing parameter size and MACs, and boosting performance.
>
> **To weakness2:** The computational cost of our model mainly comes from the dual-stage design, but several points are worth noting:
>
> 1. As the computational complexity of CSFNet, as shown in Table 2 in paper, (or reproduced below), the coarse-stage CSFNet alone already outperforms IIANet while requiring only ~1.5× its computation. This provides a clear trade-off: Users can enable the fine stage for higher performance or use only the coarse stage for lower complexity.
>
> | Method  | SI-SDRi | MACs (G) | Params (M) | Time (ms) | Memory (MB) |
> |-|-|-|-|-|-|
> | CTCNet                   | 14.3    | 167.1    | 7.0        | 84.17     | 75.80       |
> | AV-CrossNet              | 16.8    | 92.3     | 11.1       | 367.91    | 40.5        |
> | IIANet          | 16.0    | 18.6     | 3.1        | 110.11    | 12.50       |
> | CSFNet (coarse separation) | 16.9| 31.7 | 10.9   | 117.39| 42.5  |
> | CSFNet (fine separation)  | 17.5| 64.8 | 10.9  | 228.15| 83.5   |
> |
>
>
> 2. The ~3× parameter increase of CSFNet over IIANet is mainly due to IIANet’s parameter-sharing, which reduces parameter count but not proportionally its actual inference cost. Measured by GPU time, IIANet shows little advantage over CSFNet in the coarse stage, and only ~2× difference in the fine stage.
>
> 3. As shown in Table 1, Other SOTA methods like AV-CrossNet and CTCNet, despite similar SI-SDRi, do not show clear gains in efficiency and can even underperform CSFNet in computation or parameter size.
>
> 4.  We clarify that eight NVIDIA A100 (40GB) GPUs were used mainly to accelerate experiments. All benchmarks were evaluated on the same setup. However, inference does not require such high-end hardware and can run on much lower computational resources.
>
> **To question1:** Yes, several architectural choices and pretrained capabilities contribute to the substantial gains in multi-speaker scenarios beyond the coarse-to-fine process:
>
> 1. We use a multi-range BLSTM architecture (MST) to expand the receptive field, better capturing long-term dependencies in audio mixtures. As shown in Table 1 (``CSFNet (Audio-Only)''), MST significantly improves performance even in 2-speaker cases without video input, and in 3- and 4-speaker cases, the coarse stage alone (CSFNet (coarse separation)) provides noticeable gains. This design is rarely used in benchmarks.
>
> 2. Leveraging the pretrained Auto-AVSR model improves visual feature representations (lower input WER with video alone, Table 5, first column **InputWER(%)(V)**) and better aligns audio-video modalities (lower WER with both modalities, Table 5, third column **InputWER(%)(AV)**).
>
>
> **To question2:** Existing audio-visual speech separation methods follow two main approaches:
> The first is represented by RTFS-Net (Pegg et al., 2024), which handles multi-speaker mixtures by performing target speaker extraction for a single user and applying this extraction recurrently to obtain the final $n$ target. This design is reflected in the official file: nondefault_nsrc: 1: "https://github.com/spkgyk/RTFS-Net/blob/master/config". The second route separates the entire mixture jointly in a single pass, rather than extracting one speaker at a time.
> For the first, the fusion block is inherently independent of the number of speakers.
> For the second, the fusion block depends on the number of speakers, as is the case in our method.
>
> Importantly, regardless of the approach, the full model still requires training a dedicated model for each specific number of speakers. In practice, applying a model trained for $n$ speakers to mixtures containing $m \neq n$ speakers, whether $m>n$ or $m<n$, results in severely degraded performance if the goal is to separate all speakers simultaneously.
>
> Following this practice, we train separate models for different speaker counts, consistent with prior work, and this does not explain the strong results observed with more than two speakers.

---

> > ### Author Response · Authors · 2025-11-27
> >
> > Dear Reviewer,
> >
> > Regarding Weakness 1 concerning the novelty of our work, we would like to provide further clarification: While the coarse-to-fine workflow itself is a classic and widely used paradigm, the underlying perspective we adopt in the field of audio-visual speech separation is highly innovative. The detailed reasoning is as follows:
> >
> > Currently, existing audio-visual speech separation models primarily pursue performance improvements from three perspectives: 1) Optimizing the model architecture itself (e.g., AV-Crossnet, AV-ConvTasNet); 2) Leveraging powerful pretrained models (e.g., AV-HuBERT) to extract more accurate semantic information from video for auxiliary separation (e.g., AV-SepChain); 3) Developing more efficient audio-visual fusion methods (e.g., CTCNet). However, for the second perspective, even state-of-the-art pretrained video feature extractors have inherent limitations in capturing accurate semantic information within short durations (e.g., 2 seconds), reaching a performance bottleneck.
> >
> > Our work is the first in the field to combine audio and video to capture more precise and refined semantic features. Compared with video-only input, this approach significantly improves the accuracy of semantic feature extraction, which inevitably leads to substantial performance gains. From this perspective, our method proposes a new fusion paradigm that inspires how to more effectively process and utilize visual features while fully leveraging audio information. We believe this can constitute a fourth perspective for advancing audio-visual speech separation models which desires a further research in the future.
> >
> > We greatly value the opportunity to communicate with you further. With your guidance, we are confident that the quality of our manuscript will be significantly enhanced. Please do not hesitate to contact us if you have any additional questions or concerns，we will make every effort to address them promptly.
> >
> > Thank you very much for your time and consideration. We wish you all the best.
> >
> > Sincerely,
> > The Authors

---

### Author Response · Authors · 2025-11-26

Dear Reviewers,

As the discussion period for our submission, "From Coarse to Fine: Recursive Audio-Visual Semantic Enhancement for Speech Separation," draws to a close, we are writing to sincerely thank you for your insightful comments and thorough review.

I wanted to ensure we have adressed all your concerns satisfactory. If any concerns remain or if there are any final points you feel we have not fully resolved, we would be very grateful for the opportunity to provide further clarification.

Thank you again for your time and expertise, which have been invaluable in helping us improve this work.

Sincerely,

Authors

---

### Note · Authors · 2026-04-07

I have read and agree with the venue's withdrawal policy on behalf of myself and my co-authors.

---

### Meta-Review · Area_Chair_L4RX · 2026-01-03

**Summary:**

The reviewers' concerns that informed my suggested decision can be summarized as follows: (i) Lack of novelty, as the idea of coarse-to-fine processing has already been investigated in both the audio and video fields; (ii) high computational cost; (iii) performance does not scale with the computational complexity of the proposed model (the response provided by the authors is not fully convincing); (iv) lack of analysis on the specific roles of the coarse and fine stages; and (v) the proposed Speaker-wise Perceptual (SP) Fusion Block shows only minor differences compared to the fusion approach in CTCNet.

**Reviewer Concerns:**

The authors addressed some of the reviewers' concerns, for example:  (i) it was explained the element that concurred to obtain better results than other models, showing that this improvement was not  limited to the coarse-to-fine strategy. (ii) it was also clarified the proposes fusion block is more competitive in terms of complexity than the CAF Block (RTFS-Net)but a slightly reduced performance is obtained, and (iii) some new results have also been added. Nonetheless, some key concerns remain outstanding, specifically:: (i) Lack of novelty, as the idea of coarse-to-fine processing has already been investigated in both the audio and video fields; (ii) high computational cost; (iii) performance does not scale with the computational complexity of the proposed model (the response provided by the authors is not fully convincing); (iv) lack of analysis on the specific roles of the coarse and fine stages; and (v) the proposed Speaker-wise Perceptual (SP) Fusion Block shows only minor differences compared to the fusion approach in CTCNet.

**Reviewer Scores:**

Eb18 might have kept their score unchanged

28g8  might have kept their score unchanged

gVy4 might have increased their score by 1.0

WpMN would have changed their score, jumping to 8 from the initial 6 although their key concerns and questions were very plain. Accordingly, their opinion should be weighted with some caution when formulating the final decision.

---

### Decision · Program_Chairs · 2026-01-26

Reject